# Astrocytes influence medulloblastoma phenotypes and CD133 surface expression

**Emily Gronseth[1], Ankan Gupta[1], Chris Koceja[2], Suresh Kumar[3], Raman G. Kutty[4], Kevin Rarick[5], Ling Wang[6], Ramani Ramchandran** [1,6]*

1 Department of Pediatrics, Division of Neonatology, Developmental Vascular Biology Program, Children's Research Institute, Medical College of Wisconsin, Milwaukee, Wisconsin, United States of America, 2 Versiti, Milwaukee, Wisconsin, United States of America, 3 Division of Pediatric Pathology, Department of Pathology, Medical College of Wisconsin, Milwaukee, Wisconsin, United States of America, 4 Medical College of Wisconsin Affiliated Hospitals, Acsension St. Joseph Hospital, Milwaukee, Wisconsin, United States of America, 5 Department of Pediatrics, Division of Critical Care, Medical College of Wisconsin, Milwaukee, Wisconsin, United States of America, 6 Department of Obstetrics and Gynecology, Medical College of Wisconsin, Milwaukee, Wisconsin, United States of America

* rramchan@mcw.edu

**Data Availability Statement:** All relevant data are within the paper and its Supporting Information files. The microarray data is uploaded to the GEO repository per accession number GSE148540.

## Abstract

The medulloblastoma (MB) microenvironment is diverse, and cell-cell interactions within this milieu is of prime importance. Astrocytes, a major component of the microenvironment, have been shown to impact primary tumor cell phenotypes and metastasis. Based on proximity of MB cells and astrocytes in the brain microenvironment, we investigated whether astrocytes may influence MB cell phenotypes directly. Astrocyte conditioned media (ACM) increased Daoy MB cell invasion, adhesion, and *in vivo* cellular protrusion formation. ACM conditioning of MB cells also increased CD133 surface expression, a key cancer stem cell marker of MB. Additional neural stem cell markers, Nestin and Oct-4A, were also increased by ACM conditioning, as well as neurosphere formation. By knocking down *CD133* using short interfering RNA (siRNA), we showed that ACM upregulated CD133 expression in MB plays an important role in invasion, adhesion and neurosphere formation. Collectively, our data suggests that astrocytes influence MB cell phenotypes by regulating CD133 expression, a key protein with defined roles in MB tumorgenicity and survival.

## Introduction

Medulloblastoma (MB) is a pediatric brain tumor that can occur in the cerebellum or in the brainstem. Large genomic studies have stratified the tumors into at least four molecular subtypes [1, 2] which has been critical to advance research and clinical understanding of MB. Analyzing the genomic landscape of the tumor cells themselves is important, however it is now well appreciated that the tumor microenvironment has an equally important role in contributing to tumor cell fate. Most importantly, it's been shown that various factors in a tumor microenvironment have significant effects on response to therapy [3, 4]. Therefore, to improve current treatments for MB, understanding how factors and cells within the MB tumor microenvironment may be influencing tumor cells is necessary.

**Funding:** This work was supported by the Children's Research Institute and Department of Pediatrics to RR. The funder had no role in study design, data collection and analysis, decision to publish, or preparation of the manuscript.

**Competing interests:** The authors have declared that no competing interests exist.

Astrocytes are one of the most abundant cell types in the brain. In healthy conditions, these glial cells via their endfeet extensions maintain homeostasis by regulating neuronal signaling, the blood brain barrier, and neural stem cell populations. In disease states, they become 'activated' through reactive astrogliosis, which shifts their function to become immune modulating, a function shared with microglia in the brain [5]. Previously, we have shown that astrocytes influence breast cancer cell invasion and facilitates its metastasis to the brain [6, 7]. In primary brain tumors, such as MB, tumor-associated astrocytes have been found to secrete sonic hedgehog (SHH), which directly increased Nestin expression and proliferation of MB cells derived from a genetic mouse model of SHH MB [8]. Metastatic MB tumor-associated astrocytes were also recently identified to secrete chemokine C-C ligand 2 (CCL2), which enriched stem cell properties in MB cells, including expression of CD133 [9]. CD133 expression was also found to play a role in glioblastoma stem cells, wherein only CD133 positive cells showed increased invasion and radioresistance upon co-culture with astrocytes [10, 11]. Interestingly, Singh et al. [12] first isolated and described MB and glioblastoma stem-like cells using CD133 as the distinguishing marker. Here, CD133 positive cells were tumor initiating and grew as neurospheres *in vitro*, whereas CD133 negative cells lacked those properties [12]. Subsequent research on CD133's function in MB has shown that it plays a role in cell survival and therapy resistance through several pathways, including PI3K/AKT [13, 14], Notch [15], and STAT3 [9, 16]. Thus, interest in CD133 as both a target and a regulatory molecule in the MB tumor microenvironment is high. In this study, we further investigate astrocytes' influence on CD133 expression in MB cells, and the role of CD133 in contributing to ACM-associated MB phenotypes.

## Materials and methods

### Cell culture

Daoy cells were obtained from American Type Culture Collection (ATCC HTB-186). UW228/1 cells [17] were generously shared by Dr. John Silber (University of Washington). Daoy and UW228/1 cells were subject to Short Tandem Repeat (STR) analysis to confirm authentication and lack of contamination with other cells types. Both cells types were cultured in Dulbecco's Minimum Essential Medium (DMEM, Thermo Fisher Scientific, Waltham, MA) with the supplementation of 10% fetal bovine serum (FBS, Thermo Fisher Scientific) and 1% antibiotic-antimycotic (Thermo Fisher Scientific) at 37°C in a humidified 5% $CO_2$ incubator. For conditioning of MB cells, normal culture media was replaced with the respective conditioned media for the respective lengths of time.

### Preparation of conditioned media

Primary neonatal rat astrocyte conditioned media was prepared as previously described [6, 7]. For microglia conditioned media, primary neonatal rat glial cells were isolated and cultured as previously described [18]. Upon culture confluence, cells were dissociated using 0.25% Trypsin EDTA (Thermo Fisher Scientific) and microglia were isolated by fluorescence activated cell sorting (FACS) for CD11b/c positive cells. The sorted cells were cultured on poly-d-lysine coated plates (Sigma-Aldrich, 10 μg/mL) in DMEM with 10% FBS, and purity was confirmed by immunofluorescence and the absence of staining for astrocytes with a GFAP antibody. Conditioned media was collected after 48 h of incubation with microglia cultures of 70% or greater confluence. Mouse embryonic fibroblasts (MEFs) were purchased from ATCC and cultured in DMEM with 10% FBS. Conditioned media was collected after 48 h of incubation with the MEF cultures of 70% or greater confluence. All conditioned media were filtered through 0.4 μM filters prior to use to remove any dead cells or cellular debris.

## Reagents and antibodies

Anti CD133 (ab222782; 1:1000) and anti Nestin (ab134017; 1:500) antibodies were purchased from Abcam (Cambridge, MA). Anti Sox2 clone D6D9 (3579; 1:1000) and anti Oct-4A clone C30A3 (2840; 1:500) antibodies were purchased from Cell Signaling (Danvers, MA). Anti L1CAM (AF277; 1:1000) and anti-NFASC (AF2325; 1:500) were purchased from R&D Systems (Minneapolis, MN). Anti NCAM2 (sc-136328; 1:500) was purchased from Santa Cruz Biotechnology (Dallas, TX). Anti β-actin clone AC-40 (GTX11003; 1:10000) was purchased from Genetex (Irvine, CA). Anti CD11b/c clone OX-42 (201807; 1:100) was purchased from BioLegend (San Diego, CA). Anti CD133 clone TMP4 (46-1338-42: 1:200) antibody, Live/Dead fixable red stain (1:200; L34972) and Live/Dead fixable yellow stain (1:200; L34968) were purchased from Thermo Fisher Scientific. Secondary anti-rabbit IgG HRP-linked (7074; 1:10000) and anti-mouse IgG HRP-linked (7076; 1:10000) antibodies were purchased from Cell Signaling. Recombinant bFGF (233-FB) and EGF (236-EG) were purchased from R&D Systems (Minneapolis, MN).

## Microarray analysis

Cell pellets from Daoy cells cultured in their respective conditions were collected, frozen, and sent to Arraystar, Inc. (Rockville, MD) for RNA isolation and analysis via a Whole Human Genome Oligo Microarray (4 x 44K, Agilent Technologies, Santa Clara, CA). Genes with $> 2$ fold change and a $p$-value $\leq 0.05$ were considered differentially expressed. Arraystar assisted in the generation of heat map and volcano plots. Microarray data has been deposited to Gene Expression Omnibus (GEO) under accession number GSE148540.

## Real time qPCR

Total RNA was isolated using TRIzol reagent (Thermo Fisher Scientific) following the manufacturer's protocol. Total cDNA was made from 3μg RNA using SuperScriptIII reverse transcriptase (Thermo Fisher Scientific). qPCR was ran using TaqMan pre-designed probes and reagents (Thermo Fisher Scientific). The comparative Ct method was used to analyze mRNA levels using 18S as the normalization control. The following TaqMan probes were used: *L1CAM* (Hs01109748_m1); *NCAM2* (Hs00189850_m1); *NFASC* (Hs00391791_m1); *18S* (Hs99999901_s1).

## Adhesion assay

This assay was performed similar to previously described [19]. Briefly, 96-well culture plates were coated with 10 μg/mL fibronectin (Sigma-Aldrich) for 1 h at 37˚C followed by blocking with 10 mg/mL heat denatured bovine serum albumin (BSA; Rockland Immunochemicals, Inc., Limerick, PA) in PBS for 45 mins at room temperature. Plates were washed prior to use. MB cells were cultured in the respective conditions for 48 h prior to being harvested and re-seeded at 10,000 cells per well in the same media they were cultured in. Cells were seeded in quadruplicate wells and allowed to adhere for 1 h at 37˚C in a humidified 5% $CO_2$ incubator. Cells were washed thoroughly with light uniform tapping on the plates with each wash, fixed using 4% PFA, and stained with crystal violet. Brightfield images were captured using a Keyence BZ-X microscope (Osaka, Japan) at 10X magnification. Stained cells were dissolved in 30 μL 2% sodium dodecyl sulfate and optical density was assessed at 595 nm. The adherent cells were quantified either by calculating the average number of cells per image, three images per well, or by the OD 595 values.

## Boyden chamber invasion assay

Serum starved MB cells were seeded in the upper wells of 24-well Boyden chamber Matrigel coated invasion inserts (BD Biosciences). The inserts were placed in wells with normal or conditioned media and incubated at 37°C in a humidified 5% $CO_2$ incubator for the indicated time points. The inserts were then fixed in 4% PFA for 15 mins at room temperature, and then stained with crystal violet (Sigma). The cells remaining in the upper well were removed with a cotton swab. The invasive cells were imaged using both a Zeiss Axio Observer Z1 (Oberkocken, Germany) at 10X magnification and the Keyence BZ-X microscope (Osaka, Japan), also at 10X magnification. The invasive cells were quantified by calculating the average number of cells per image, five images per insert.

## Zebrafish xenotransplant MB cell injections

Zebrafish work was performed under approved AUA320 protocol by the MCW IACUC committee. We utilized 48 hours post fertilization (hpf) embryos of the *Tg(flk:mCherry)* reporter line which has red fluorescent vasculature crossed on an *Absolut*$^{+/+}$ background (*ednrbl*$^{-/--}$ *mitfa*$^{-/-}$) to enable complete transparency through adulthood. Prior to injection, embryos were manually dechorionated, screened for vascular mCherry expression, and anesthetized with tricaine. Daoy cells stably expressing m-Emerald were cultured for 48 h in control media or ACM. Cells were collected and brought to a single cell suspension in PBS + 5% FBS at a concentration of 50–100 cells in 1–4 nL (optimized previously by dispensing the injected volume into oil on a microscope slide and manually counting). Cells were then loaded into a pulled glass microneedle broken to approximately 15 μm internal and 18 μm external diameters. The microneedle was driven by a WPI pv820 Pneumatic Picopump. Cells were microinjected into the visible hindbrain ventricle of the embryos, positioned using a soft agarose mold in alignment with the injection apparatus. Two hours after injection, embryos to be imaged were anesthetized and laterally mounted in 1% agarose on glass bottom dishes, with E2 buffer covering the top.

## Zebrafish xenotransplant confocal imaging and quantification

A LSM510 Meta confocal microscope (Carl Zeiss, Jena, Germany) was used for the time-lapse imaging of the xenotransplanted cells. The microscope was programmed to take images every three mins for 24 h, spanning the entire depth of cells injected. The imaging chamber was kept humidified and at 32°C (temperature pre-determined to keep both embryos and cells alive) with 5% $CO_2$ [20]. To analyze cell movement and morphology of the time lapse images, Volocity software (PerkinElmer, Waltham, MA) was used. Individual cells were tracked manually, and the positions were then connected to determine the movement and path of each visible cell. Additionally, a reference point on the vasculature of the fish was manually tracked to account for any movement of the embryo during the time lapse, which was then subtracted from the calculation movement of the MB cells. For every image, the relative location/center of each cell was recorded for X, Y, and Z planes. To calculate total distance moved of each cell, the absolute change in location was calculated from each previous position and the sums from the entire time-lapse calculated (with any change in distance/location of the reference point subtracted at every position, prior to taking the sum). Total displacement was calculated by taking the square root of the sums of movement in X, Y, and Z planes. Not all cells remained within the visible field for the entirety of the 24 h time-lapse, therefore total distance and displacement were calculated for the first 250 frames, or about 12.5 h. To analyze cell morphology, cellular protrusions were measured. At every 20 frames, the diameter of each visible cell and the length of any protrusions were measured using Volocity software.

## Flow cytometry analysis

Cells were trypsinized, collected, and stained with anti-CD133 antibody as per manufacturer's recommendation, for 30 mins at 4˚C. Dead cell exclusion stain was also included and thus CD133 expression was analyzed only on viable cells. Samples were acquired on LSR Fortessa X20 (BD) and data was analyzed using FlowJo software (Treestar, Ashland, OR).

## Neurosphere formation assay

MB cells were brought to a single cell suspension in serum free media. For neurosphere formation cultures, cells were seeded at $0.5 \times 10^6$ cells/mL on low-attachment plates in their control or experimental media, in the presence of 20 ng/mL bFGF and 20 ng/mL EGF. Serum free ACM was collected by culturing astrocytes in DMEM (high glucose, no glutamine, no phenol red, Thermo Fisher Scientific) and collecting the supernatant after 48 h. Control media is DMEM alone, and as a positive control condition for sphere formation, DMEM was mixed 1:1 with Ham's F12 nutrient mix (Thermo Fisher Scientific). At 48 h, cells were imaged using a Keyence BZ-X microscope. In a six-well plate, 20 pre-programmed brightfield images per well were taken at 4X magnification with 1.7X digital zoom applied. The images were analyzed using Keyence analysis software, where sphere diameter was measured. To quantify, all spheres greater than 75 μm in diameter were measured and counted. Representative images used for visualization of the neurospheres were taken at 10X magnification and 1.8X digital zoom was applied. Measurement of neurosphere size was also done previously using a rough calibration of pixels to micrometer (μm), and then used ImageJ for further data processing. Both methods (Keyence software analysis and ImageJ analysis) yielded similar data.

## siRNA knockdowns

Trilencer-27 short-interfering RNAs (siRNAs) targeting human *CD133* were pre-designed by and purchased from Origene (SR305838; Rockville, MD). The sequence of *siCD133-1* is 5′- GGCAGAUAGCAAUUUCAAGGACUTG-3′; and the sequence of *siCD133-2* is 5′-GGCUUGGA AUUAUGAAUUGCCUGCA-3′. Gene knockdown was validated by qRT-PCR and western blotting. siRNAs were transfected into Daoy cells according to the manufacturer's protocol. Briefly, Daoy cells were seeded to be 50–70% confluent that the time of transfection. Per well of a 6-well plate, a complex of 400 μL Opti-MEM (Thermo Fisher Scientific), 1 μL Lipofectamine RNAiMax (Thermo Fisher Scientific), and *CD133* siRNA (10 nM) or Scramble siRNA (10 nM) were added to the cells for 24–48 h.

## Protein isolation and immunoblotting

Cells cultured for protein isolation were washed with PBS prior to adding cold RIPA buffer (Thermo Fischer Scientific) supplemented with protease (cOmplete EDTA-free Protease Inhibitor cocktail, Roche, Basal, Switzerland) and phosphatase (PhoseSTOP, Roche) inhibitors. Lysates were centrifuged at 16000xg for 20 mins at 4˚C followed by the collection of supernatant. Protein quantification was determined by DC Protein Assay (Bio-Rad Laboratories, Hercules, CA). Total protein lysates were loaded onto 4–20% SDS-PAGE gels (Mini-PROTEAN TGX Precast Protein Gels, Bio-Rad Laboratories) and transferred to nitrocellulose membrane. Blots were blocked at room temperature for 1 h in 5% BSA in TBST prior to incubation with primary antibody at 4˚C overnight. After washing, blots were incubated with HRP-linked secondary antibodies for 1 h at room temp. Chemiluminescent signal was detected using enhanced chemiluminescent (ECL) HRP substrate (SuperSignal West Pico and Fempto solutions, Thermo Fisher Scientific) and imaged using a ChemiDoc Imaging System

(Bio-Rad Laboratories). Quantification of optical densitometry was performed using ImageJ software.

## Statistical analysis

All experiments performed had a minimum of three biological replicates. Statistical analysis was done using GraphPad Prism software (GraphPad Software, Inc., San Diego, CA). Data were presented as the mean ± standard deviation (SD) calculated by two-tailed Student's *t*-test, one-way analysis of variance (ANOVA), or two-way ANOVA, as appropriate. For all analyses, a p-value ≤ 0.05 was considered statistically significant.

## Results

### Astrocyte secreted factors alter MB cell gene expression

To investigate astrocytes effect on MB tumor cells, we performed a hypothesis-generating microarray-based gene expression analysis on MB cells post culturing in ACM. In the past, culturing breast cancer cells in ACM has shown profound gene expression changes [6, 7]. ACM was used in place of culture media for Daoy cells for 48 hours (h) ("ACM conditioned", S1A Fig). Gene expression was quantified by microarray, and the results are shown through a hierarchical clustering heat map (Fig 1A) and volcano plot (Fig 1B). When comparing ACM-conditioned MB cells to control cultured MB cells, there were a total of 973 differentially expressed genes, 756 of which were greater than two-fold differentially expressed ($p \leq 0.05$) (Fig 1B, red squares). Gene expression pathway analysis was performed on all genes greater than two-fold differentially expressed ($p \leq 0.05$) in ACM-cultured cells in comparison to control. Of note, functional mapping of genes to Kyoto Encyclopedia of Genes and Genomes (KEGG) pathways found an enrichment in genes related to cell adhesion upregulated in ACM-treated MB cells. Adhesion-related networks were also identified using Ingenuity Pathway Analysis software (Qiagen) (S1B Fig). We also validated three microarray targets (*NFASC*, *L1CAM*, *NCAM2*) related to adhesion network, which were up in ACM-treated MB cells at both mRNA (S1C Fig) and proteins levels (S1D Fig). These results demonstrated the breadth of effects of astrocyte-secreted factors on MB cells, and pointed to adhesion-related phenotypes.

### Astrocyte-secreted factors increase invasion and adhesion of MB cells

To confirm the up regulated cell adhesion gene expression results, we functionally tested ACM-treated MB cells on a two-dimensional adhesion plate assay coated with fibronectin. Here, Daoy and UW228/1 cells were cultured in ACM or control DMEM for 48 h prior to reseeding for 1 h in fresh culture media. Daoy cells cultured in ACM adhered more strongly to the fibronectin-coated plates compared to control cultured Daoy cells (Fig 1C). UW228/1 cells showed no difference in adhesion between ACM-cultured and control cultured cells (S2A Fig). We next assessed cell invasion based on our past data where ACM culturing increased invasion of breast cancer cells (MDA-MB-231), lung cancer adenocarcinoma cells (H2030) and sarcoma cells (S180) [6]. Using an *in vitro* Boyden chamber invasion assay, we assessed whether ACM has an effect on the invasion of MB cells. Daoy and UW228/1 cells demonstrated increased invasion with ACM in the lower chamber compared to all control conditions: DMEM, DMEM + 10% FBS, DMEM + 5% FBS, and DMEM SF (Serum Free) (Fig 1D and 1G, S2B Fig). Additionally, ACM SF in the lower chamber induced invasion of Daoy cells similar to that of ACM with serum, demonstrating Daoy cell invasion in ACM conditions was serum-independent (Fig 1D and 1G). UW228/1 cell showed less invasion in response to ACM SF

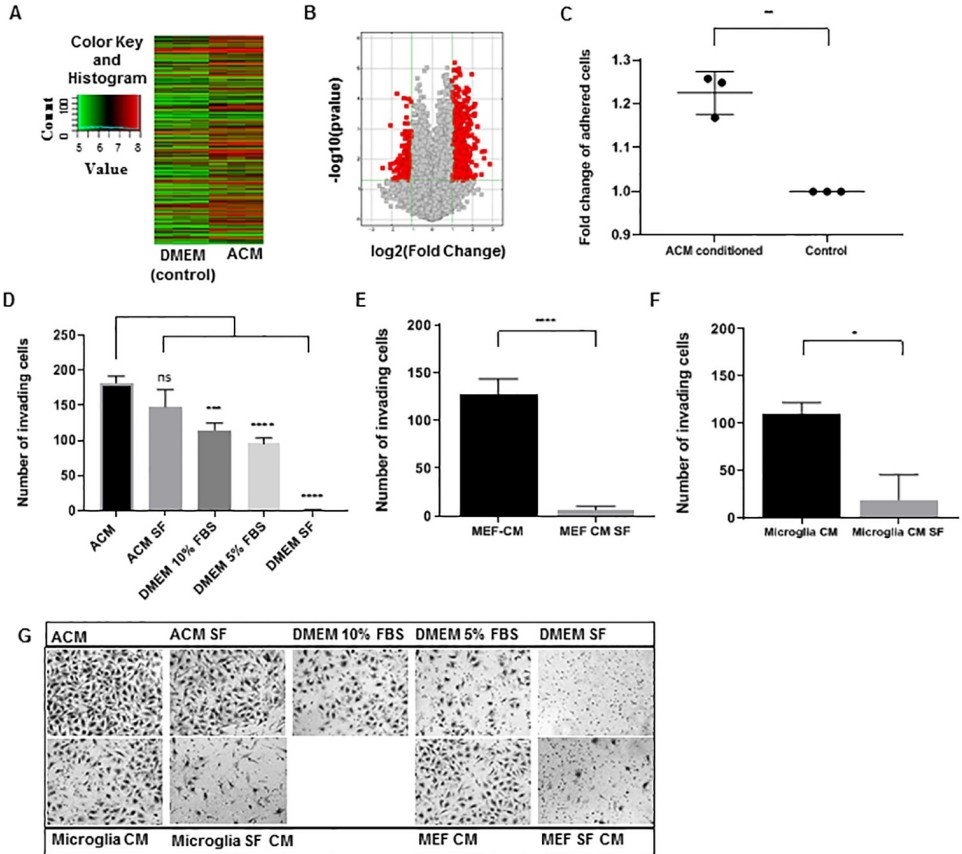

**Fig 1. Astrocyte secreted factors alter Daoy MB cell gene expression, adhesion and invasion.** (A) Hierarchical clustering heat map of differentially expressed genes in Astrocyte Conditioned Media (ACM) cultured cells compared to DMEM cultured (control) cells. Image modified from Arraystar (Rockville, MD) (B) Volcano plot of differentially expressed genes between ACM cultured and control cells. The *y*-axis represents the–log10 of the p-value while the *x*-axis represents the log2 of gene expression fold change. All differentially expressed genes are shown as squares, with the red squares representing the differentially expressed genes (≥2 fold up or down) with statistical significance (p≤0.05) in ACM cultured cells. Image modified from Arraystar (Rockville, MD) (C) Fold change of the number of adhered cells per image in ACM conditioned cells compared to control (DMEM cultured, set to 1). (D-E) Average number of invaded Daoy cells per image, counted on the lower surface of Boyden chambers, with the respective media in the lower well during incubation. (F) Representative images of invaded cells on the lower surface of the Boyden chambers. Quantified values expressed as mean +/- SD. *p<0.05, **p<0.01, ***p<0.001, ****p<0.0001.

compared to normal ACM, however ACM still induced significantly greater invasion compared to all DMEM controls (S2B Fig).

To determine if this effect on invasion was astrocyte specific or a common response to conditioned culture media from other cells, we also assayed Daoy and UW228/1 cell invasion using conditioned media (normal and SF) from other cell types. We isolated, cultured, and collected conditioned media from neonatal rat microglia. Daoy and UW228/1 cells showed invasion with microglia conditioned media in the lower chamber, however invasion significantly decreased with microglia SF conditioned media (Fig 1E and 1G, S2C–S2E Fig). To test conditioned media from another highly proliferative cell type, mouse embryonic fibroblasts (MEF) cells were cultured and conditioned media was collected. Similarly, Daoy and UW228/1 cell showed substantial invasion in response to MEF conditioned media, however this invasion was largely ablated when serum was not present (Fig 1F and 1G, S2D–S2E Fig).

Taken together, these data demonstrate that, in addition to the MB cell adhesion gene signature, astrocyte-secreted factors present in ACM, induce phenotypic changes that increase adhesion to fibronectin and increase chemotactic invasion through matrigel. Further, the invasion response is specific to conditioned media from astrocytes, and the invasion is for the most part, serum-independent.

## MB cells conditioned in ACM prior to xenograft show increased cellular protrusion formation

Next, we wanted to investigate whether these phenotypic changes seen *in vitro* also translated to changes in cell behavior *in vivo*. Recent methodologies have established zebrafish as an effective vertebrate system to visualize xenografted tumor cell behavior and movement [21]. Specifically, pediatric brain tumors have been successfully xenografted in zebrafish for drug screening purposes [22, 23]. A zebrafish xenotransplant model combined with confocal time-lapse imaging were used to determine the behavior of astrocyte conditioned MB cells *in vivo*. Because of the transparency of the embryonic zebrafish model, it provides the ability to not only visualize cell migration, but also cell morphology changes as related to the changes in adhesion and invasion seen *in vitro*. In addition, the zebrafish adaptive immune system is not developed until 3–4 weeks post fertilization (wpf) [16, 24, 25], therefore the short-term imaging within 24 h of injection bypassed the need to immunocompromise the animals chemically, genetically or by γ-irradiation. Daoy cells stably expressing mEmerald (variant of GFP) protein were generated and subsequently cultured in either ACM or control media for 48 h and micro-injected into the hindbrains of 48 hours post fertilization (hpf) embryos. The approximate MB cell location and appearance post microinjection is marked (Fig 2A). Injected zebrafish embryos were then laterally mounted in agarose and imaged via time-lapse confocal microscopy for 24 h (S1 Video). Cells were analyzed by tracking their movement (Fig 2B) and measuring protrusions (Fig 2C) using Volocity software (PerkinElmer). Overall, total distance, displacement, and average displacement per frame were unchanged by culturing Daoy cells in ACM prior to injection (Fig 2D–2F). Interestingly, ACM conditioning distinctly changed MB cell morphology by increasing the overall occurrence (Fig 2G) and size of cellular protrusions (Fig 2H) of MB cells in the zebrafish brain. Thus, ACM-conditioned MB tumor cells do possess phenotypic activity *in vivo* that resembles the observed *in vitro* adhesive phenotype.

## Astrocyte-secreted factors increase CD133 surface expression and neurosphere characteristics in MB cells

The hypothesis from the microarray array and subsequent confirmation by adhesion assay *in vitro* that ACM-conditioned MB cells showed more adhesion suggested that molecules involved in adhesion and MB cell invasion will be functionally relevant on ACM-conditioned MB cells. It is known that Daoy MB cells cultured as neurospheres express higher CD133 (a pentaspanin transmembrane glycoprotein that has been associated with protrusion formation and localization [26]) when compared to Daoy cells cultured in an adherent monolayer [27, 28]. Further, these neurosphere-forming MB cells were more invasive *in vitro* [27], and CD133 expression is high in disseminated human MB tumors [9]. Thus, we investigated CD133 surface expression in ACM-conditioned MB cells compared to control-media conditioned MB cells using flow cytometry. We also analyzed CD15 expression, another protein which has been shown to be expressed on MB stem-like cells [29], specifically in the SHH subtype. Daoy cells were cultured for 24, 48, and 72 h in either ACM or control media prior to analysis. ACM culturing increased CD133 surface expression 2-fold at 24 and 48 h, and close to 3-fold at 72 h in comparison to control cultured Daoy cells (Fig 3A). We did not observe an effect of ACM

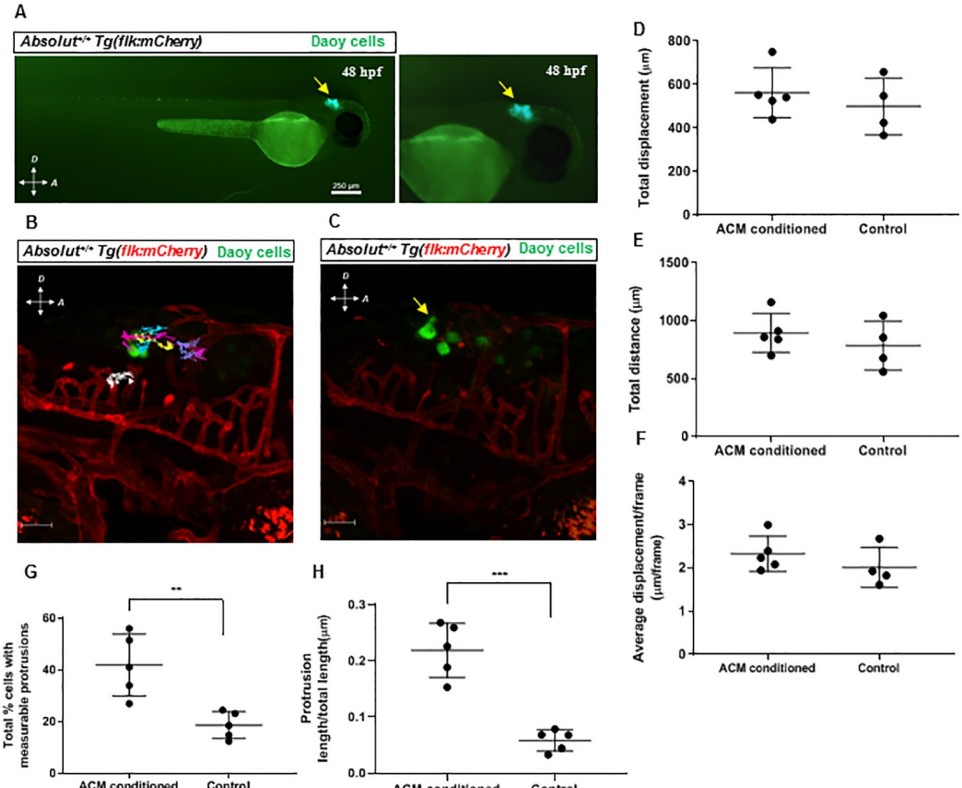

**Fig 2. ACM conditioning changes Daoy MB cell morphology *in vivo*.** (A) Representative images of cells (yellow arrow) after injection into the hindbrains of 48 hpf *Absolut^{+/+} Tg(flk:mCherry)* zebrafish embryos. Right panel is higher magnification. Scale bar is 250 μm. (B) Representative image of injected cells at the beginning of the time-lapse imaging. Colored lines show the tracked movement of individual cells during the time-lapse imaging, and the white line shows the tracked reference point movement. Scale bar is 50 μm. (C) Representative image of injected cell protrusion (yellow arrow). Scale bar is 50 μm. (D-F) Quantification of injected cell migration, represented by total cell displacement (D) and total distance (E) moved by all cells in first 12.5 hours (h) of imaging, and average displacement/ frame (F) of cells through entire 24 h time lapse. (G-H) Quantification of injected Daoy cell protrusion formation, measured by average percent of visible cells with protrusions (G) and protrusion length to cell body size ratio (H). Quantified values expressed as mean +/- SD. **p<0.01, ***p<0.001.

culturing on CD15 expression at any time point (Fig 3B). We next assessed whether ACM culturing enhances neurosphere formation of Daoy cells. Daoy cells were dissociated into a single cell suspension and seeded in ACM, DMEM, or positive control DMEM:F12 media in neurosphere culturing conditions (all SF, low attachment culture dishes supplemented with bFGF and EGF). At 48 h, brightfield images were taken and quantified by the number of neurospheres imaged as well as the diameter of the neurospheres. Culturing in DMEM alone produced almost no neurosphere formation (Fig 3C and 3D). However, ACM and DMEM:F12 conditions induced robust neurosphere formation at 48 h, where the size and number of neurospheres in ACM conditions were similar to that of the positive control (Fig 3B and 3C). Two different methods of neurosphere quantification were performed as explained in the materials and methods section. Next, we probed Daoy neurosphere lysates for neural stem cell markers Nestin, Oct-4A and Sox2 [30–32]. Nestin, an intermediate filament protein, was increased in ACM and DMEM:F12 cultured neurospheres, compared to DMEM alone (Fig 3E). These results are consistent with previous reports that show high Nestin expression in CD133 positive MB stem-like cells [12, 15, 28]. Oct-4A is a transcription factor which has been associated with poor outcomes in MB patients, in addition to increasing neurosphere formation and

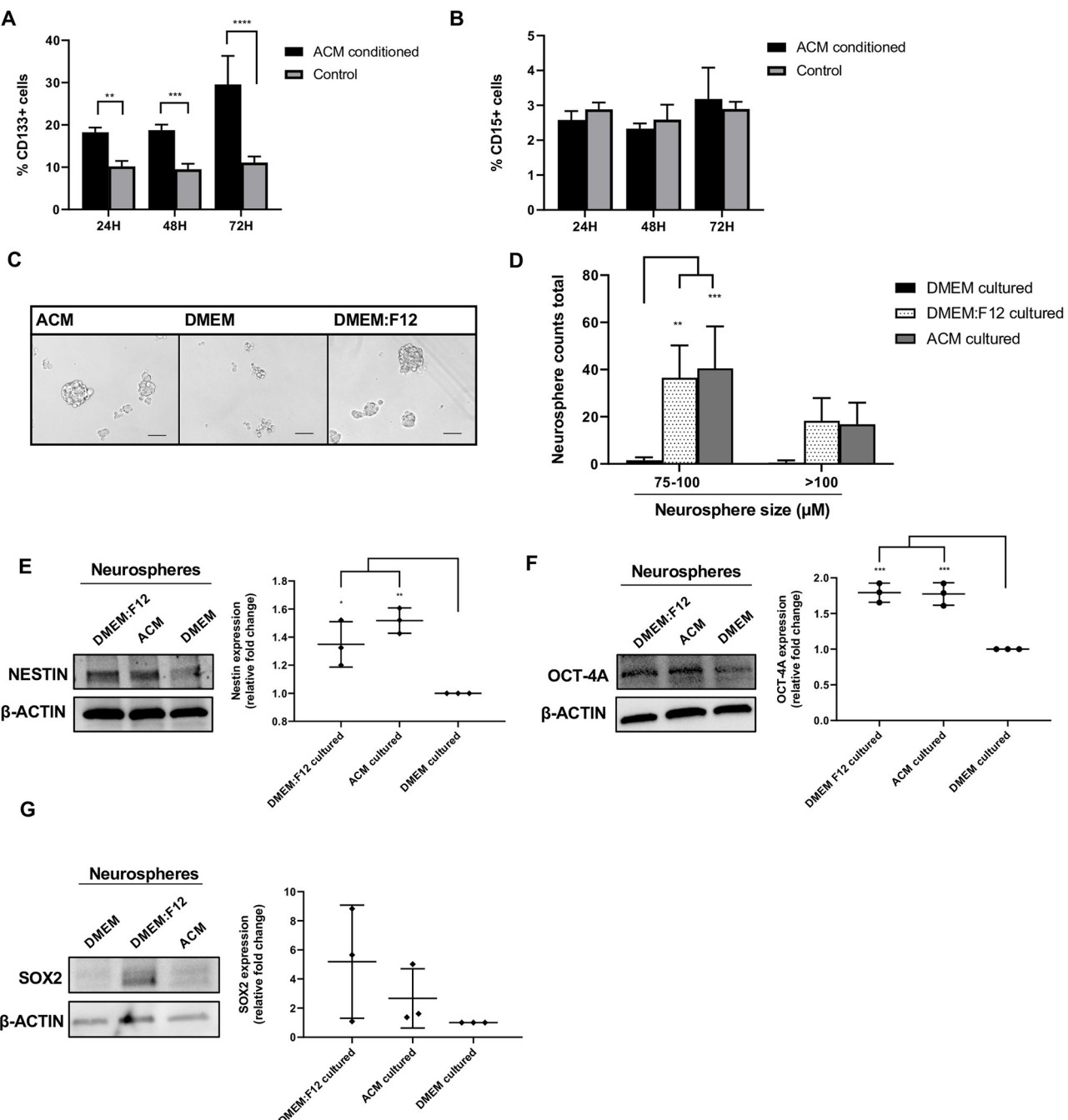

**Fig 3. ACM effects on CD133 surface expression and neurosphere formation.** (A) Percentage of CD133 positive cells cultured in ACM or DMEM (control), measured by flow cytometry at 24, 48, and 72 h. (B) Percentage of CD15 positive cells cultured in ACM or DMEM (control), measured by flow cytometry at 24, 48, and 72 h. (C) Representative images of Daoy cells cultured in neurosphere conditions using the respective media for 48 h. Scale bar is 100 μm. (D) Total number of neurospheres counted in all images between 75–100 μm or >100 μm at 48 h, cultured in the respective media. (E) Western blot analysis of Nestin, (F) Oct-4A and (G) Sox2 expression in respective neurosphere culturing conditions at 48 h. β-actin served as a loading control (left). Quantification of relative fold change of Nestin, Oct-4A and Sox2 expression, measured by the densitometry of western blot bands and normalized to β-actin (DMEM condition set to 1) (right). Quantified values expressed as mean +/- SD. *p<0.05, **p<0.01, ***p<0.001, ****p<0.0001.

invasion upon overexpression in Daoy cells [33, 34]. Here, Daoy neurospheres cultured in DMEM:F12 and ACM conditions expressed higher Oct-4A compared to control cultured neurospheres (Fig 3F). A third neural stem cell transcription factor, Sox2 [32], showed no difference in expression between DMEM and ACM-cultured neurospheres, and extreme variability in DMEM:F12 neurospheres (Fig 3G) Together, these data demonstrate that ACM conditioning of MB cells increases CD133 surface expression, Nestin expression, Oct-4A expression and neurosphere formation in Daoy cells.

## Knockdown of CD133 in MB cells reduces adhesion, invasion, and neurosphere formation

To determine the relevance of increased CD133 surface expression on ACM-MB cells, we utilized short interfering RNA (siRNA) to knockdown *CD133* expression, followed by functional assessment of *CD133*'s role in adhesion, invasion, and neurosphere formation. Two separate 27-mer siRNAs (*siCD133-1* and *siCD133-2*) targeting different regions of *CD133* mRNA were utilized, and knockdown efficiency was assessed by quantifying gene and protein expression after transfection with *siCD133-1*, *siCD133-2*, *scramble* siRNA, as well as non-transfected cells. CD133 protein was effectively knocked down by *siCD133-1* and *siCD133-2* by 66% and 80%, respectively (Fig 4A).

Daoy cells were transfected prior to being cultured in either ACM or control media for 48 h before being reseeded in the adhesion assay described in the methods. When cultured in ACM, *siCD133-1* and *siCD133-2* transfected cells were significantly less adhesive than *scramble* siRNA transfected cells (Fig 4B). Interestingly, in control cultured cells, *CD133* knockdown had no effect on adhesion (Fig 4B), suggesting that CD133 may work together with another factor in ACM to increase *in vitro* adhesion.

Similarly, *CD133* was knocked down in Daoy cells prior to seeding the cells in the upper well of the Boyden invasion chamber, with either ACM or DMEM +10% FBS in the lower well. Overall, *CD133* knockdown cells showed significantly less invasion with both ACM and DMEM in the lower well in comparison to scramble transfected cells (Fig 4C). With ACM in the lower well, invasion decreased in *siCD133-1* and *siCD133-2* transfected MB cells by a combined average of 46% compared to *scramble* siRNA transfected cells. With DMEM in the lower well, *siCD133-1* and *siCD133-2* transfected cells showed about 61% less invasion compared to *scramble* siRNA transfected cells (Fig 4C). We also assessed neurosphere formation (DMEM, ACM and DMEM:F12 neurosphere conditions) with the *CD133* knockdown MB cells. As observed before, non-transfected cells cultured in DMEM alone showed very little neurosphere formation, which remained unchanged in *CD133* knockdown and *scramble* siRNA conditions (Fig 4D). In both ACM and DMEM:F12 culture conditions, neurosphere formation was significantly reduced in both size and number in *CD133* knockdown cells compared to *scramble* siRNA transfected cells (Fig 4D).

The above results suggest that upon ACM conditioning, the observed increase in CD133 surface expression is associated with the displayed MB phenotypes of increased adhesion, invasion, and neurosphere formation.

## Discussion

The MB microenvironment is comprised of various cell types, and depending on where the tumor is located, cells such as astrocytes, microglia, and others (neuronal cells, endothelial cells, pericytes, neural stem cells, ependymal cells and connective tissue that make up the meningeal layers) [35] can interact with MB cells. Our study focuses on astrocytes and their role in the MB microenvironment. Salient features of this study include: (a) Astrocytes' influence MB

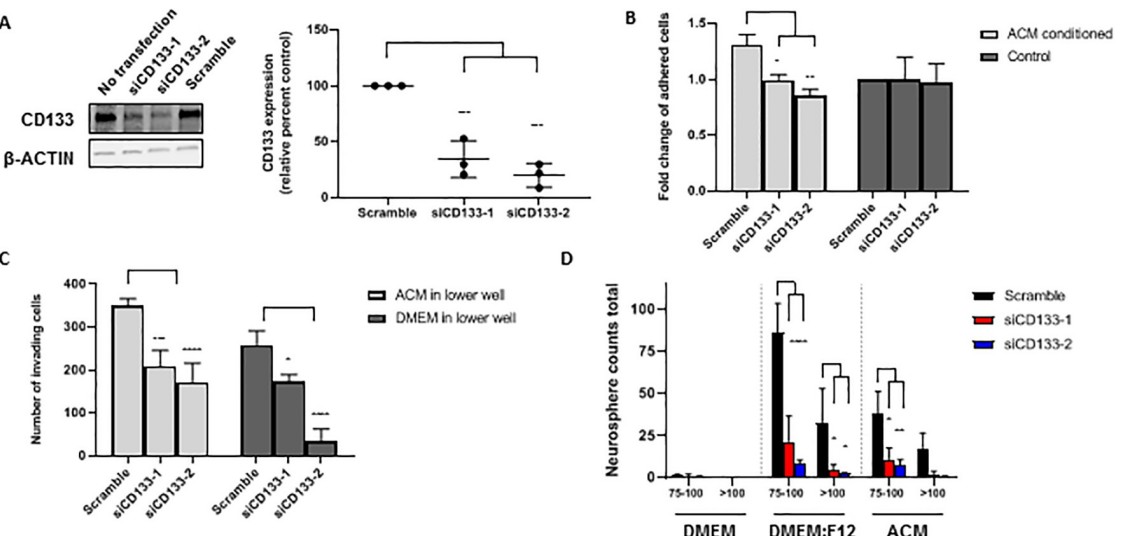

**Fig 4. Short interfering RNA (siRNA) knockdown of *CD133* alters Daoy MB cell function.** (A) Western blot analysis of CD133 expression 48 h after transfection with *siCD133-1*, *siCD133-2*, scramble siRNA, or no transfection. β-actin served as a loading control (left). Quantification of relative percent CD133 expression, measured by the densitometry of western blot bands and normalized to β-actin (Scramble set to 100%) (right). (B) Fold change of the number of adhered cells per image after transfecting with *siCD133-1*, *siCD133-2* or scramble siRNA, then culturing in ACM or DMEM (control) (DMEM scramble set to 1). (C) Average number of invading cells per image, counted on the lower surface of Boyden chambers with ACM or DMEM in the lower well, after transfecting with *siCD133-1*, *siCD133-2* or scramble siRNA. (D) Total number of neurospheres counted with diameter length between 75–100 μm or >100 μm. Cells were transfected with *siCD133-1*, *siCD133-2* or scramble siRNA and then cultured in the respective media for 48 h. Quantified values expressed as mean +/- SD. $^{*}p < 0.05$, $^{**}p < 0.01$, $^{***}p < 0.001$, $^{****}p < 0.0001$.

cells through up regulation of CD133 expression, a protein known to be important for MB tumorigenicity and survival. (b) The ACM-mediated increase in MB CD133 expression plays a role in MB cell invasion, adhesion, and neurosphere formation.

To investigate astrocytes' role in the MB microenvironment, we used ACM collected from primary astrocyte cultures to assess its effect on MB cell function. With this model, we can observe paracrine signaling from factors secreted by astrocytes and downstream effects, versus the effects of physical interactions. We found that culturing human MB cells in ACM increased adhesion to fibronectin in Daoy cells, as well as invasion in both Daoy and UW228/1 cells. Previous research from our group found ACM to have a similar effect on invasion of other tumor cell types, including breast cancer cells [6, 7]. Moreover, those studies showed that culturing MDA-MB-231 breast cancer cells in ACM led to an increase in brain metastases in a mouse xenograft model. Interestingly, when we cultured MB cells in ACM prior to a zebrafish xeno-transplant, we did not see a difference in cell migration during the 24 h imaging time frame; however, long-term metastasis could not be assessed in this model. We did observe increased MB cell protrusion formation in the zebrafish brain, which may be related to the increased adhesion or CD133 expression seen *in vitro*, as there is evidence in the literature for both [26, 36], however this was not investigated here.

The ACM-induced CD133 expression was indeed proven to be functionally significant. Knockdown of *CD133* reduced neurosphere formation in the positive control neurosphere condition (DMEM:F12), which is consistent with previous studies that have demonstrated enhanced neurosphere formation in FACS-isolated CD133 positive Daoy cells [28, 37]. CD133 also plays a role in ACM-induced neurosphere formation. Knockdown of *CD133* reduced invasion of Daoy MB cells altogether, with both ACM and DMEM in the lower well. CD133's role in MB invasion has not been well defined to this point, limited to correlations with

increased expression and invasion or metastasis of MB cells [38–40]. One such correlative study compared Daoy adherent and neurosphere cultures and found higher *CD133* expression and migration in neurosphere cells [27]. Our research corroborates this correlation by providing direct evidence that CD133 influences Daoy invasion. To date, there has been little work investigating the role of CD133 in cell adhesion. Notably, in ovarian cancer cells, overexpression of *CD133* increased adhesion to mesothelial layers and collagen [41]. We investigated CD133's role in adhesion of MB cells. Interestingly, knockdown of *CD133* only reduced adhesion in ACM culturing conditions, however control conditions remained unaffected. This suggests CD133's role in adhesion is dependent on paracrine signaling from astrocytes and may require a yet unknown factor in ACM or downstream signaling event caused by ACM culturing to exert its functional effect on adhesion.

Understanding which factors in ACM that contribute to increasing CD133 expression will increase our mechanistic understanding of how ACM-induced effects may translate into MB outcomes and suggest avenues for intervention. The astrocyte secretome is diverse and thus, there are likely a combination of factors in the ACM influencing the MB phenotypes. CCL2, SHH, and IL-4 are astrocyte-secreted factors which have been shown to promote MB phenotypes [8, 9, 42]. Interestingly, CCL2 secretion from astrocytes [9] was induced by chemotherapy and radiation-induced inflammation and necroptosis in astrocytes. Moreover, the IL-4 secretion was found to come from astrocytes which trans-differentiated from the MB precursor cells in a SHH MB mouse model [42]. These reports combined with our findings provide supporting evidence that astrocytes influence the MB microenvironment in distinct ways, which must be considered during the development of anti-MB treatment strategies. As of now, it is not clear what astrocyte factor(s) is responsible for inducing CD133 expression in MB cells in our model. This is a topic of future investigation.

To summarize, this study provides a direct link between an important survival and stem cell protein, CD133, and *in vitro* phenotypes that correlate to poor outcomes in MB malignancy. We also demonstrate that CD133 surface expression is directly affected by paracrine signaling from astrocytes, neighboring cells in the MB microenvironment. Our work and the work of others are increasingly suggestive that CD133 is an important cell surface target in MB, given that CD133 is known for its role in survival and resistance of MB cancer stem cell populations [43]. Thus, these questions form the basis of future investigations. Lastly, our studies demonstrate the importance of studying astrocytes and their role in the MB microenvironment to influence MB malignancy.

## Supporting information

**S1 Fig. Schematic of ACM collection and use in culturing of MB cells, and microarray validation of select adhesion targets.** (A) depicts the ACM collection and treatment of MB cells in a pictorial format. (B) Graphic depiction of adhesion pathways predicted to be altered in ACM cultured Daoy cells based on the microarray data. Image generated using Ingenuity Pathway Analysis (Qiagen) software. (C) qRT-PCR normalized fold change compared to control (DMEM, set to 1) for three adhesion target genes (*NFASC*, *L1CAM* and *NCAM2*) identified as significantly changed in the microarray analysis. (D) Western blot analysis of NFASC, L1CAM, and NCAM2 in the ACM-conditioned and DMEM-conditioned MB cell lysates. β-actin served as a loading control (left). Quantification of relative fold change measured by the densitometry of western blot bands and normalized to β-actin (DMEM condition set to 1) (right). Quantified values expressed as mean +/- SD. *p<0.05, **p<0.01, ***p<0.001, ****p<0.0001.
(PDF)

**S2 Fig. Conditioned media (CM) effects on adhesion and invasion in UW228/1 cells.** (A-B) Fold change of the number of adhered cells per well, measured by dissolving the stained cells and reading optical density at 595 nm in ACM conditioned cells compared to control (DMEM, set to 1). (B-D) Average number of invaded cells per image, counted on the lower surface of Boyden chambers, with the respective media in the lower well during incubation. Panel C is microglia conditioned media (CM) cells and (D) is mouse embryonic fibroblast (MEF) media conditioning. (E) Representative images of invaded cells on the lower surface of the Boyden chambers in response to various conditioned media are shown. Quantified values expressed as mean +/- SD. $^{**}p<0.01$, $^{***}p<0.001$, $^{****}p<0.0001$.
(TIF)

**S1 Video. Representative zebrafish xenotransplant time-lapse imaging video.** Daoy cells (green) were either cultured in ACM or control (DMEM) media prior to injecting into *Absolut$^{+/+}$ Tg(flk:mCherry)* 48 hpf zebrafish embryos (vasculature is red). Confocal z-stack images were taken every three minutes for 24 h. Images were rendered into a 3-dimensional video using Volocity software.
(AVI)

**S1 Raw images.**
(PDF)

# Acknowledgments

The authors thank Children's Research Institute (CRI) Imaging Core facility for training and guidance in time-lapse confocal microscopy. We thank Dr. John Silber (University of Washington, Seattle, WA) for generously providing the UW228 cell lines, and Dr. Tamjid Chowdhury for his input on experiments for this project. Also, thanks to Noah Leigh and Dr. Shubhangi Prabhudesai for maintaining the zebrafish facility at Medical College of Wisconsin.

# Author Contributions

**Conceptualization:** Ling Wang, Ramani Ramchandran.

**Data curation:** Raman G. Kutty.

**Funding acquisition:** Ramani Ramchandran.

**Investigation:** Emily Gronseth, Ankan Gupta, Chris Koceja.

**Methodology:** Emily Gronseth, Ankan Gupta, Chris Koceja, Suresh Kumar, Raman G. Kutty, Kevin Rarick, Ling Wang.

**Resources:** Kevin Rarick, Ramani Ramchandran.

**Software:** Suresh Kumar.

**Supervision:** Ramani Ramchandran.

**Validation:** Emily Gronseth.

**Visualization:** Chris Koceja, Suresh Kumar.

**Writing – original draft:** Emily Gronseth.

**Writing – review & editing:** Emily Gronseth, Ankan Gupta, Suresh Kumar, Kevin Rarick, Ling Wang, Ramani Ramchandran.

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
