## [Decision Letter · Decision Letter 0]

26 Mar 2020

PONE-D-20-06629

Astrocytes influence medulloblastoma phenotypes and CD133 surface expression

PLOS ONE

Dear Dr. Ramchandran,

Thank you for submitting your manuscript to PLOS ONE. After careful consideration, we feel that it has merit but does not fully meet PLOS ONE’s publication criteria as it currently stands. Therefore, we invite you to submit a revised version of the manuscript that addresses the points raised during the review process.

Albeit of interest, the paper in its present form requires to be consistently amended in several parts, following all the points highlighted by the two referees. Actually to perform a microarray does not make sense if the altered genes are not investigated and without a confirmation by RT-PCR. Moreover, the stemness must be observed by means of the specific genes including at least Nanog and OCT3/4. Discussion is largely speculative and long. Therefore, the Authors must amend each point raised in the referee's comments in their revised paper.

We would appreciate receiving your revised manuscript by May 10 2020 11:59PM. To enhance the reproducibility of your results, we recommend that if applicable you deposit your laboratory protocols in protocols.io, where a protocol can be assigned its own identifier (DOI) such that it can be cited independently in the future. For instructions see: http://journals.plos.org/plosone/s/submission-guidelines#loc-laboratory-protocols

We look forward to receiving your revised manuscript.

Kind regards,

Gianpaolo Papaccio, M.D., Ph.D.

Academic Editor

PLOS ONE

Journal Requirements:

2. Thank you for including your ethics statement:  AUA 320 for zebrafish animal study. No human studies in this project.

3. Please amend your current ethics statement to include the full name of the ethics committee that approved your specific study.

For additional information about PLOS ONE submissions requirements for ethics oversight of animal work, please refer to http://journals.plos.org/plosone/s/submission-guidelines#loc-animal-research. Once you have amended this/these statement(s) in the Methods section of the manuscript, please add the same text to the “Ethics Statement” field of the submission form (via “Edit Submission”).

4. We note that you are reporting an analysis of a microarray, next-generation sequencing, or deep sequencing data set. PLOS requires that authors comply with field-specific standards for preparation, recording, and deposition of data in repositories appropriate to their field. Please upload these data to a stable, public repository (such as ArrayExpress, Gene Expression Omnibus (GEO), DNA Data Bank of Japan (DDBJ), NCBI GenBank, NCBI Sequence Read Archive, or EMBL Nucleotide Sequence Database (ENA)). In your revised cover letter, please provide the relevant accession numbers that may be used to access these data. For a full list of recommended repositories, see http://journals.plos.org/plosone/s/data-availability#loc-omics or http://journals.plos.org/plosone/s/data-availability#loc-sequencing.

5. In your Methods section, please provide additional details regarding the cell lines used in your study. For the UW228/1 cell line, please address whether the cell line was verified, and if so, how it was verified. Please also ensure that you have cited the original paper describing this cell line. For the Daoy cell line, please include the ATCC catalog number. For more information on PLOS ONE's guidelines for research using cell lines, see https://journals.plos.org/plosone/s/submission-guidelines#loc-cell-lines.

Reviewers' comments:

Reviewer's Responses to Questions

**Comments to the Author**

1. Is the manuscript technically sound, and do the data support the conclusions?

Reviewer #1: Yes

Reviewer #2: Partly

2. Has the statistical analysis been performed appropriately and rigorously? 

Reviewer #1: Yes

Reviewer #2: I Don't Know

3. Have the authors made all data underlying the findings in their manuscript fully available?

Reviewer #1: Yes

Reviewer #2: Yes

4. Is the manuscript presented in an intelligible fashion and written in standard English?

Reviewer #1: No

Reviewer #2: Yes

5. Review Comments to the Author

Reviewer #1: The manuscript is interesting and highlights the role of astrocytes in supporting medulloblastoma tumorigenesis. Although this, there are some concerns that need to be addressed. Following microarray analyses, the authors should confirm the main genes involved in adhesion by real time PCR. Moreover, they should indicate also whether there are specific pathways downregulated.

The image of neurospheres must be changed. The authors must add the pictures with an original magnification grater in order to detect the differences in size following treatments. Moreover, to confirm the stemness of cells treated with ACM, the authors must evaluate stem transcriptional factors such as OCT4, Nanog and Sox2. The authors demonstrate that CD133 is associated with increased adhesion. As the microarray indicates that adhesion genes are increased, the authors should analyze the genes with the highest value for each experimental condition and evaluate how they change. When CD133 is silenced, what do it happens to these genes?

The Discussion section must be revised. It is too speculative. It should be better focused on the importance of results obtained in medulloblastoma treatment.

Reviewer #2: In this manuscript the Authors investigate about the effect of astrocytes conditioned media (ACM) on medulloblastoma cell in terms of adhesion, invasion and stemness. They performed basic studies in vitro showing that ACM have an effect on the adhesion, migration and spheroids formation of MB cells. Moreover they use a zebrafish model to show the effect of ACM in an in vivo model. They also knocked down CD113 to show its role in the ACM induced effects. The work is interesting in principle, however there are few concerns that need to be addressed as follows:

It is not clear why the Authors performed a microarray, showing a strong alteration of adhesion gene, but did not investigate the role of any of those genes in MB to establish a mechanist effect.

The Author state that ACM increased the stemness of MB cells, however this was based only on spheroid formation. For instance the spheroid analysis and images are not acceptable, more clear and higher magnification images should be provided and an algorithm to calculate sphere diameter should be used. Then, to show an increase in stemness features only spheres formation it’s not enough, the Author should at least show an increase of stem-associated-genes such as Nanog, OCT3/4, and Sox-2 or perform side population assay, or ALDH or any other assay appropriated for MB. This is valid as well for the CD133 knock down.

Discussion should be more focused and less speculative.

6. PLOS authors have the option to publish the peer review history of their article (what does this mean?). If published, this will include your full peer review and any attached files.

Reviewer #1: No

Reviewer #2: No

---

## [Author Response · Author response to Decision Letter 0]

15 Apr 2020

REVIEWER RESPONSE LETTER

Re: PONE-D-20-06629: Astrocytes influence medulloblastoma phenotypes and CD133 surface expression

Editor Comments:

Albeit of interest, the paper in its present form requires to be consistently amended in several parts, following all the points highlighted by the two referees. Actually to perform a microarray does not make sense if the altered genes are not investigated and without a confirmation by RT-PCR. 

Response: We appreciate the concern here. The microarray was performed as a hypothesis-generating experiment. Based on this experiment, we observed that adhesion pathways (Fig. S1B) were influenced in medulloblastoma tumor cells that were conditioned in astrocyte media. Yes, we did validate the targets. In the revision, we have now included the qPCR (Fig. S1C) and western blot (Fig. S1D) results for select microarray targets related to adhesion (NFASC, L1CAM, NCAM2) that were upregulated in ACM-MB cells. 

Moreover, the stemness must be observed by means of the specific genes including at least Nanog and OCT3/4. 

Response: We performed Sox2 western blot (Fig. 3F), which is also a stem marker and found no difference between ACM-conditioned and DMEM-conditioned MB cells. Due to current COVID-19 situation, we are unable to perform other stem marker genes as requested. We have therefore made revisions in the text to de-emphasize the stemness point.

Discussion is largely speculative and long. Therefore, the Authors must amend each point raised in the referee's comments in their revised paper.

Response: We have made extensive changes to the discussion to make the points that are directly relevant to the data in the manuscript, and those that support or oppose existing literature on this subject. We provide below a point-by-point response to the reviewer queries. 

Journal Requirements:

 Response: We have ensured that the manuscript and its file names meet PLOS ONE style requirements.

3. Please amend your current ethics statement to include the full name of the ethics committee that approved your specific study.

For additional information about PLOS ONE submissions requirements for ethics oversight of animal work, please refer to http://journals.plos.org/plosone/s/submission-guidelines#loc-animal-research. Once you have amended this/these statement(s) in the Methods section of the manuscript, please add the same text to the “Ethics Statement” field of the submission form (via “Edit Submission”).

Response: The requested changes have been made.

4. We note that you are reporting an analysis of a microarray, next-generation sequencing, or deep sequencing data set. PLOS requires that authors comply with field-specific standards for preparation, recording, and deposition of data in repositories appropriate to their field. Please upload these data to a stable, public repository (such as ArrayExpress, Gene Expression Omnibus (GEO), DNA Data Bank of Japan (DDBJ), NCBI GenBank, NCBI Sequence Read Archive, or EMBL Nucleotide Sequence Database (ENA)). In your revised cover letter, please provide the relevant accession numbers that may be used to access these data. For a full list of recommended repositories, see http://journals.plos.org/plosone/s/data-availability#loc-omics or http://journals.plos.org/plosone/s/data-availability#loc-sequencing.

Response: We plan to submit the microarray data to the GEO database and will provide the reference number upon completion.

5. In your Methods section, please provide additional details regarding the cell lines used in your study. For the UW228/1 cell line, please address whether the cell line was verified, and if so, how it was verified. Please also ensure that you have cited the original paper describing this cell line. For the Daoy cell line, please include the ATCC catalog number. For more information on PLOS ONE's guidelines for research using cell lines, see https://journals.plos.org/plosone/s/submission-guidelines#loc-cell-lines.

Response: Short Tandem Repeat (STR) genetic profiling for cell authentication was performed on UW228/1 cells. Although these cells do not have a public STR profile for positive comparison, the STR analysis confirmed no match with any other cell lines within the DSMZ database in addition to confirming a single donor source. This is noted in the revised manuscript.

Response: The requested supporting information is included.

Response: This information has been provided in the supporting information (file “S1_raw_images”).

 

Reviewers' comments:

We thank both reviewers for evaluating our manuscript and providing constructive comments. 

Reviewer #1: 

The image of neurospheres must be changed. The authors must add the pictures with an original magnification grater in order to detect the differences in size following treatments. 

Response: Yes, we realize this concern but due to COVID-19 related lab closures, hibernations, and other issues, it is unclear when institution will open up. Deliveries are also delayed these days, and thus the May 10 deadline is not feasible. Thus, we are unable to perform additional new experiments to generate spheres for better high power imaging. For better visualization of the spheres, we enlarged the representative images (Fig 3) as much as possible while maintaining good image quality. In terms of size determinations, we have used two independent methods of quantifications for images taken on two different scopes (Zeiss and Keyence), which has been explained in detailed later (rev. 2 query below), and showed the same result. We also excluded spheres less than 75 �m to avoid including aggregates in our quantification. Thus, we are confident with our neurosphere quantification. 

Moreover, to confirm the stemness of cells treated with ACM, the authors must evaluate stem transcriptional factors such as OCT4, Nanog and Sox2. 

Response: We performed Sox2 western blot (Fig. 3F), which is also a stem marker and found no difference between ACM-conditioned and DMEM-conditioned MB cells. Due to current COVID-19 situation, we are unable to perform other stem marker genes as requested. We have therefore made revisions in the text to de-emphasize the stemness point.

The authors demonstrate that CD133 is associated with increased adhesion. As the microarray indicates that adhesion genes are increased, the authors should analyze the genes with the highest value for each experimental condition and evaluate how they change. When CD133 is silenced, what do it happens to these genes?

Response: The microarray was performed as a hypothesis-generating experiment. Based on this experiment, we observed that adhesion pathways (Fig. S1B) were influenced in medulloblastoma tumor cells (Daoy) that were conditioned in astrocyte media. Yes, we did validate the targets. In the revision, we have now included the qPCR (Fig. S1C) and western blot (Fig. S1D) results for select adhesion gene targets (NFASC, L1CAM, NCAM2) upregulated in ACM-MB cells. We never claimed that the adhesion gene targets NFASC, L1CAM and NCAM2 were downstream of CD133, and thus did not see the rationale for performing this experiment.

The Discussion section must be revised. It is too speculative. It should be better focused on the importance of results obtained in medulloblastoma treatment.

Response: We have made extensive changes to the discussion to make the points that are directly relevant to the data in the manuscript, and those that support or oppose existing literature on this subject. Please see revised manuscript with track changes.

Reviewer #2: 

It is not clear why the Authors performed a microarray, showing a strong alteration of adhesion gene, but did not investigate the role of any of those genes in MB to establish a mechanist effect.

Response: We apologize if the microarray rationale was not clear. The microarray was performed as a hypothesis-generating experiment. Based on this experiment, we observed that adhesion pathways (Fig. S1B) were influenced in medulloblastoma (MB) tumor cells that were conditioned in astrocyte media. Yes, we did validate the adhesion gene targets. In the revision, we have now incuded the qPCR (Fig. S1C) and western blot (Fig. S1D) results for select targets (NFASC, L1CAM, NCAM2) upregulated in ACM-MB cells. We also performed knockdown of L1CAM target but did not observe changes in adhesion (data not shown). Thus, we rationalized redundancy in the up regulated adhesion molecules, and focused on a known CD133 adhesion molecule that is associated with MB progression [1]. Without the microarray data, we would not have hypothesized that the adhesion pathway was involved in astrocyte conditioned MB cell phenotype. Some of these explanations have been included in the revised MS. 

The Author state that ACM increased the stemness of MB cells, however this was based only on spheroid formation. For instance the spheroid analysis and images are not acceptable, more clear and higher magnification images should be provided and an algorithm to calculate sphere diameter should be used. 

Response: We have now de-emphasized the stemness point, because our Sox2 western blot data showed great variability between DMEM-conditioned MB cells and ACM-conditioned MB cells (Fig. 3F), which precludes any conclusion. We actually performed two independent methods of quantification for images taken under two different scopes. We initially took pictures on Zeiss scope. To measure diameter, we performed a rough calibration of pixels to micrometer (�m), and then used ImageJ for further data processing. In the second Keyence scope, we pre-programmed set points for picture, which removes bias, and used the Keyence analysis software to directly measure micrometer size of the spheres. In this second method, the diameter measurement was more straightforward, and unbiased. Further, we only calculated spheres with greater than 75 �m to avoid including aggregates. Results from both methods showed the same result. In terms of new and better images, we will need to perform the experiment again. Under normal circumstances, we would be more than happy to do it. But, unfortunately, due to COVID-19 associated lab and institution closure, we cannot perform new experiments. For the representative images in Figure 3, we have enlarged our existing pictures (as much as possible while maintaining good image quality) for better visualization of the spheres. 

Then, to show an increase in stemness features only spheres formation it’s not enough, the Author should at least show an increase of stem-associated-genes such as Nanog, OCT3/4, and Sox-2 or perform side population assay, or ALDH or any other assay appropriated for MB. This is valid as well for the CD133 knock down.

Response: We appreciate the reviewer suggestion, and did perform Sox2 western blot (Fig. 3F), which is also a stem marker [2] and found no difference between ACM-conditioned and DMEM-conditioned MB cells. Due to current COVID-19 situation, we are unable to perform other stem marker genes as requested. We have therefore made revisions in the text to de-emphasize the stemness point.

Discussion should be more focused and less speculative.

Response: We have made extensive changes to the discussion to make the points that are directly relevant to the data in the manuscript, and those that support or oppose existing literature on this subject.

We thank the reviewers for spending their valuable time providing us feedback on our manuscript.

REFERENCES

1. Singh SK, Clarke ID, Terasaki M, Bonn VE, Hawkins C, Squire J, et al. Identification of a cancer stem cell in human brain tumors. Cancer Res. 2003;63(18):5821-8. PubMed PMID: 14522905.

2. Ellis P, Fagan BM, Magness ST, Hutton S, Taranova O, Hayashi S, et al. SOX2, a persistent marker for multipotential neural stem cells derived from embryonic stem cells, the embryo or the adult. Dev Neurosci. 2004;26(2-4):148-65. PubMed PMID: 15711057.

---

## [Decision Letter · Decision Letter 1]

5 May 2020

PONE-D-20-06629R1

Astrocytes influence medulloblastoma phenotypes and CD133 surface expression

PLOS ONE

Dear Dr. Ramchandran,

Thank you for submitting your manuscript to PLOS ONE. After careful consideration, we feel that it has merit but does not fully meet PLOS ONE’s publication criteria as it currently stands. Therefore, we invite you to submit a revised version of the manuscript that addresses the points raised during the review process.

The Authors only partly positively responded to the comments previously raised by the two referees. In their answer the Authors wrote that they were unable to perform extra experiments due to the SARS-COV-2 pandemia. This Editor understand that the problem is real but asks the Authors to obtain much more time for research completion. Therefore The Authors are allowed to ask for more time to send their revised manuscript.

We would appreciate receiving your revised manuscript by Jun 19 2020 11:59PM. To enhance the reproducibility of your results, we recommend that if applicable you deposit your laboratory protocols in protocols.io, where a protocol can be assigned its own identifier (DOI) such that it can be cited independently in the future. For instructions see: http://journals.plos.org/plosone/s/submission-guidelines#loc-laboratory-protocols

We look forward to receiving your revised manuscript.

Kind regards,

Gianpaolo Papaccio, M.D., Ph.D.

Academic Editor

PLOS ONE

Reviewers' comments:

Reviewer's Responses to Questions

**Comments to the Author**

1. If the authors have adequately addressed your comments raised in a previous round of review and you feel that this manuscript is now acceptable for publication, you may indicate that here to bypass the “Comments to the Author” section, enter your conflict of interest statement in the “Confidential to Editor” section, and submit your "Accept" recommendation.

Reviewer #1: (No Response)

Reviewer #2: (No Response)

2. Is the manuscript technically sound, and do the data support the conclusions?

Reviewer #1: Yes

Reviewer #2: Partly

3. Has the statistical analysis been performed appropriately and rigorously? 

Reviewer #1: Yes

Reviewer #2: I Don't Know

4. Have the authors made all data underlying the findings in their manuscript fully available?

Reviewer #1: Yes

Reviewer #2: Yes

5. Is the manuscript presented in an intelligible fashion and written in standard English?

Reviewer #1: No

Reviewer #2: Yes

6. Review Comments to the Author

Reviewer #1: The Authors do not fully address my previous concerns. They state that, due to current COVID-19 situation, they are unable to perform experiments requested by this reviewer. I recommend asking to editor more time to carry out the experiments in order to improve the manuscript.

Reviewer #2: the author have addressed part of the comments raised by this reviewer, accoridng to author's responce they cannot perform required experiment because of the covid-19 spread and lab closure. Due to this situation I believ it will be more approriate a time extention so the revisin can be perform at the best.

7. PLOS authors have the option to publish the peer review history of their article (what does this mean?). If published, this will include your full peer review and any attached files.

Reviewer #1: No

Reviewer #2: No

---

## [Author Response · Author response to Decision Letter 1]

10 Jun 2020

REVIEWER RESPONSE LETTER

Re: PONE-D-20-06629: Astrocytes influence medulloblastoma phenotypes and CD133 surface expression

Editor Comments: The Authors only partly positively responded to the comments previously raised by the two referees. In their answer the Authors wrote that they were unable to perform extra experiments due to the SARS-COV-2 pandemia. This Editor understand that the problem is real but asks the Authors to obtain much more time for research completion. Therefore, the authors are allowed to ask for more time to send their revised manuscript.

Reviewer #1: The Authors do not fully address my previous concerns. They state that, due to current COVID-19 situation, they are unable to perform experiments requested by this reviewer. I recommend asking to editor more time to carry out the experiments in order to improve the manuscript.

Reviewer #2: the author have addressed part of the comments raised by this reviewer, according to author's responce they cannot perform required experiment because of the covid-19 spread and lab closure. Due to this situation I believe it will be more appropriate a time extension so the revising can be performed at the best.

Response: Thank you for the additional time that the editor provided to perform additional experiments and make more revisions to the manuscript. As per the reviewer and editor requests, we have now performed additional experiments and included new data in the following panels Figs. 3C, 3D, and 3F.

New images for neurospheres were requested. We performed a new experiment to generate newer images under higher power to honor this request. All of our previous images were low power. This new higher magnified image data is now included in figure 3C with scale bars. The neurosphere sizes are clearly larger in the astrocyte-conditioned media (ACM) compared to DMEM conditions. Further, the ACM-induced neurospheres are similar in size to DMEM:F12 positive control conditions. The new images were quantified, and the quantification is also provided in the figure 3D.

Western blots for additional stem cell proteins (Nanog & Oct4) were requested in addition to previously provided Nestin (Fig. 3E) and Sox2 (Fig. 3G) data. We performed westerns for Oct-4 and Nanog, and have provided new data for Oct4 (Fig. 3F) along with quantification. The Oct-4A blot shows that under both ACM and DMEM:F12 positive control conditions, Oct-4A protein (***p<0.001) expression was higher compared to DMEM conditions. We checked Nanog protein as well, but the antibody did not work well, and thus no conclusion can be further derived from this blot. Collectively, of the three stemness proteins that were detected in MB cells, we noticed 2 (Nestin and Oct-4A) out of 3 expression to be higher in ACM conditions compared to DMEM conditions. 

Stemness write-up de-emphasis. We carefully searched for the entire manuscript for the stemness or stem cell word, and only found that we utilized these words in reference to describing CD133 as a stem cell marker or describing Nestin, Sox2 or Oct-4A proteins in the results section. The Discussion section was extensively revised in the previous submission, and all references to stemness was removed then.

---

## [Decision Letter · Decision Letter 2]

24 Jun 2020

Astrocytes influence medulloblastoma phenotypes and CD133 surface expression

PONE-D-20-06629R2

Dear Dr. Ramchandran,

We’re pleased to inform you that your manuscript has been judged scientifically suitable for publication and will be formally accepted for publication once it meets all outstanding technical requirements.

Kind regards,

Gianpaolo Papaccio, M.D., Ph.D.

Academic Editor

PLOS ONE

Additional Editor Comments (optional):

Reviewers' comments:

Reviewer's Responses to Questions

**Comments to the Author**

1. If the authors have adequately addressed your comments raised in a previous round of review and you feel that this manuscript is now acceptable for publication, you may indicate that here to bypass the “Comments to the Author” section, enter your conflict of interest statement in the “Confidential to Editor” section, and submit your "Accept" recommendation.

Reviewer #1: All comments have been addressed

Reviewer #2: All comments have been addressed

2. Is the manuscript technically sound, and do the data support the conclusions?

Reviewer #1: Yes

Reviewer #2: Yes

3. Has the statistical analysis been performed appropriately and rigorously? 

Reviewer #1: Yes

Reviewer #2: I Don't Know

4. Have the authors made all data underlying the findings in their manuscript fully available?

Reviewer #1: Yes

Reviewer #2: Yes

5. Is the manuscript presented in an intelligible fashion and written in standard English?

Reviewer #1: Yes

Reviewer #2: Yes

6. Review Comments to the Author

Reviewer #1: (No Response)

Reviewer #2: The Authors have adressed al the concerns raised by this reviewer. Now the manuscript is greatly improved.

7. PLOS authors have the option to publish the peer review history of their article (what does this mean?). If published, this will include your full peer review and any attached files.

Reviewer #1: No

Reviewer #2: No

---

## [Editor Report · Acceptance letter]

25 Jun 2020

PONE-D-20-06629R2 

Astrocytes influence medulloblastoma phenotypes and CD133 surface expression 

Dear Dr. Ramchandran:

I'm pleased to inform you that your manuscript has been deemed suitable for publication in PLOS ONE. Congratulations! Your manuscript is now with our production department. 

Kind regards, 

on behalf of

Prof. Gianpaolo Papaccio 

Academic Editor

PLOS ONE